# Uncertainty Sampling is Preconditioned Stochastic Gradient Descent on Zero-One Loss

**Stephen Mussmann**
Department of Computer Science
Stanford University
Stanford, CA
mussmann@stanford.edu

**Percy Liang**
Department of Computer Science
Stanford University
Stanford, CA
pliang@cs.stanford.edu

## Abstract

Uncertainty sampling, a popular active learning algorithm, is used to reduce the amount of data required to learn a classifier, but it has been observed in practice to converge to different parameters depending on the initialization and sometimes to even better parameters than standard training on all the data. In this work, we give a theoretical explanation of this phenomenon, showing that uncertainty sampling on a convex (e.g., logistic) loss can be interpreted as performing a preconditioned stochastic gradient step on the population zero-one loss. Experiments on synthetic and real datasets support this connection.

## 1 Introduction

Active learning algorithms aim to learn parameters with less data by querying labels adaptively. However, since such algorithms change the sampling distribution, they can introduce bias in the learned parameters. While there has been some work to understand this (Schütze *et al.*, 2006; Bach, 2007; Dasgupta and Hsu, 2008; Beygelzimer *et al.*, 2009), the most common algorithm, "uncertainty sampling" (Lewis and Gale, 1994; Settles, 2010), remains elusive. One of the oddities of uncertainty sampling is that sometimes the bias is *helpful*: uncertainty sampling with a subset of the data can yield lower error than random sampling on *all* the data (Schohn and Cohn, 2000; Bordes *et al.*, 2005; Chang *et al.*, 2017). But sometimes, uncertainty sampling can vastly underperform, and in general, different initializations can yield different parameters asymptotically. Despite the wealth of theory on active learning (Balcan *et al.*, 2006; Hanneke *et al.*, 2014), a theoretical account of uncertainty sampling is lacking.

In this paper, we characterize the dynamics of a variant of uncertainty sampling to explain the bias introduced. For convex models, we show that uncertainty sampling with respect to a *convex loss on all the points* is performing a preconditioned [1] *stochastic gradient step on the (non-convex) population zero-one loss*. Furthermore, each uncertainty sampling iterate in expectation moves in a descent direction of the zero-one loss, unless the parameters are at an approximate stationary point. This explains why uncertainty sampling sometimes achieves lower zero-one loss than random sampling, since that is the quantity it implicitly optimizes. At the same time, as the zero-one loss is non-convex, we can get stuck in a local minimum with higher zero-one loss (see Figure 1).

Empirically, we validate the properties of uncertainty sampling on a simple synthetic dataset for intuition as well as 25 real-world datasets. Our new connection between uncertainty sampling and the zero-one loss minimization clarifies the importance of a good (sufficiently large) seed set, rather than

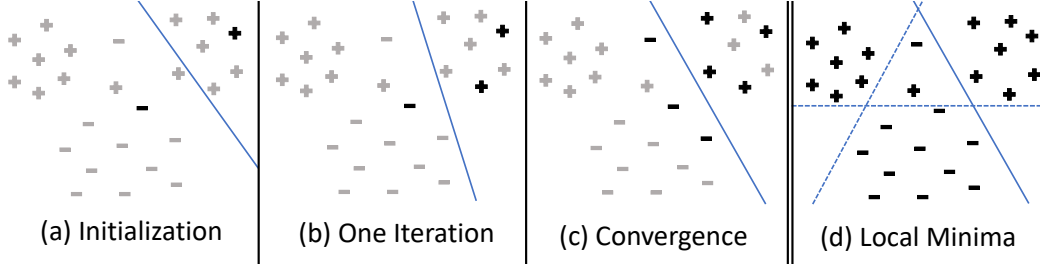

| (a) Initialization | (b) One Iteration | (c) Convergence | (d) Local Minima |

Figure 1: A typical run of uncertainty sampling. Each iteration, uncertainty sampling chooses to label the point closest to the current decision boundary. (a) Random initialization of uncertainty sampling. (b) A point closest to the decision boundary is added. (c) Several more points around the decision boundary are added until convergence. We see that uncertainty sampling uses only a fraction of the data, but converges to a local minimum of the zero-one loss. (d) Three different local minima of the zero-one loss are shown, where the horizontal linear classifier much more preferable to the other two.

using a single point per class, as is commonly done in the literature (Tong and Koller, 2001; Yang and Loog, 2016).

## 2  Setup

We focus on binary classification. Let $z = (x, y)$ be a data point, where $x \in \mathbb{R}^k$ is the input and $y \in \{-1, 1\}$ is the label, drawn from some unknown true data distribution $z \sim p^*$. Assume we have a family of scoring functions $S(x, \theta)$, where $\theta \in \mathbb{R}^d$ are the parameters; for linear models, we have $S(x, \theta) = \theta \cdot \phi(x)$, where $\phi : \mathbb{R}^k \to \mathbb{R}^d$ is the feature map.

Given parameters $\theta$, we predict 1 if $S(x, \theta) > 0$ and $-1$ otherwise, and therefore err when $y$ and $S(x, \theta)$ have opposite signs. Define $Z(\theta)$ to be the zero-one loss (misclassification rate) over the data distribution, generally the quantity of interest:

$$Z(\theta) \stackrel{\text{def}}{=} \mathbb{E}_{(x,y) \sim p^*}[H(-yS(x, \theta))], \tag{1}$$

where $H$ is the Heaviside function:

$$H(x) \stackrel{\text{def}}{=} \begin{cases} 0 & x < 0, \\ \frac{1}{2} & x = 0, \\ 1 & x > 0. \end{cases} \tag{2}$$

Note that the *training* zero-one loss is step-wise constant, and the gradient is 0 almost everywhere. However, if the PDF of $p^*$ is continuous, then the *population* zero-one loss is differentiable at most parameters, a fact that will be shown later.

Since minimizing the zero-one loss is computationally intractable (Feldman *et al.*, 2012), it is common to define a convex surrogate $\ell((x, y), \theta) = \psi(yS(x, \theta))$ which upper bounds the zero-one loss; for example, the logistic loss is $\psi(s) = \log(1 + e^{-s})$. Given a labeled dataset $\mathcal{D} = \{z_1, \ldots, z_n\}$, we can define the estimator that minimizes the sum of the loss plus quadratic regularization:

$$\theta_{\mathcal{D}} \stackrel{\text{def}}{=} \arg \min_{\theta} \sum_{z \in \mathcal{D}} \ell(z, \theta) + \lambda \|\theta\|_2^2, \tag{3}$$

This can generally be solved efficiently via convex optimization.

**Passive learning: random sampling.**  Define the population loss as

$$L(\theta) \stackrel{\text{def}}{=} \mathbb{E}_{z \sim p^*}[\ell(z, \theta)]. \tag{4}$$

In standard passive learning, we sample $\mathcal{D}$ randomly from the population and compute $\theta_{\mathcal{D}}$. As $|\mathcal{D}| \to \infty$, the parameters will generally converge to the minimizer of $L$ (note this is in general distinct from the minimizer of $Z$).

**Active learning: uncertainty sampling.** In active learning, we have access to a pool of $n_{\text{pool}}$ unlabeled data points (known $x$, unknown $y$) drawn from $p^*$ and adaptively choose the points to label. In this work, we analyze uncertainty sampling (Lewis and Gale, 1994; Settles, 2010), which is widely used for its simplicity and efficacy (Yang and Loog, 2016).

Let us denote our label budget as $n$, the number of points we label. Uncertainty sampling (Algorithm 1) begins with $n_{\text{seed}} < n$ labeled points $\mathcal{D}$ drawn randomly from the pool and minimizes the regularized loss (3) to obtain initial parameters. Then, the algorithm draws a random minipool (subset $\mathcal{X}_{\text{M}}$ of the data pool $\mathcal{X}_{\text{U}}$), and chooses the point $x \in \mathcal{X}_{\text{M}}$ that the current model is most uncertain about, i.e., the one with the smallest absolute value of score.[2] It then queries $x$ to get the corresponding label $y$ and adds $(x, y)$ to $\mathcal{D}$. Finally, we update the model by optimizing (3). A key difference between this version of uncertainty sampling and most other versions is that we remove the minipool $\mathcal{X}_{\text{M}}$ from $\mathcal{X}_{\text{U}}$ after choosing a point from it; this is done simply for theoretical convenience. The process is continued until we have labeled $n$ points in total.

---

**Algorithm 1** Uncertainty Sampling

---

**Input:** Loss $\ell$, regularization parameter $\lambda$, label budget $n$, labeled $\mathcal{D}$ of size $n_{\text{seed}}$, unlabeled $\mathcal{X}_{\text{U}}$ of size $n_{\text{pool}}$, minipool size $n_{\text{minipool}}$
Train $\theta_{n_{\text{seed}}} = \arg\min_\theta \sum_{z \in \mathcal{D}} \ell(z, \theta) + \lambda \|\theta\|_2^2$
**for** $t = (n_{\text{seed}} + 1), \ldots, n$ **do**
    Draw a random subset $\mathcal{X}_{\text{M}}$ of size $n_{\text{minipool}}$ from $\mathcal{X}_{\text{U}}$
    Choose $x = \arg\min_{x \in \mathcal{X}_{\text{M}}} |S(x, \theta)|$
    Query $x$ to get label $y$
    $\mathcal{D} = \mathcal{D} \cup \{(x, y)\}$
    $\mathcal{X}_{\text{U}} = \mathcal{X}_{\text{U}} \setminus \mathcal{X}_{\text{M}}$
    Train $\theta_t = \arg\min_\theta \sum_{z \in \mathcal{D}} \ell(z, \theta) + \lambda \|\theta\|_2^2$
**end for**
Return $\theta_n$

---

We have four hyperparameters related to the number of data points: $n_{\text{seed}}$, $n_{\text{minipool}}$, $n_{\text{pool}}$, and $n$. We start with $n_{\text{seed}}$ labeled points and $n_{\text{pool}}$ unlabeled points, and select a new point from a random subset of size $n_{\text{minipool}}$ sampled without replacement until we have $n$ labeled points in total. Note that $n_{\text{pool}} \geq n \cdot n_{\text{minipool}}$.

## 3 Theory

We present three results shedding light on uncertainty sampling that build on each other. First, in Section 3.1, we show how the optimal parameters change with the addition of a single point to the convex surrogate (e.g. logistic) loss. Then, we show that uncertainty sampling is preconditioned stochastic gradient descent on the zero-one loss in Section 3.2. Finally, we show that uncertainty sampling iterates in expectation move in a descent direction of $Z$ in Section 3.3.

### 3.1 Incremental Parameter Updates

First, we analyze how the sample convex surrogate loss minimizer changes with each additional point; these are the iterates of uncertainty sampling. Let us assume the loss is convex and thrice-differentiable with bounded derivatives:

**Assumption 1** (Convex Surrogate Loss). *The surrogate loss $\ell(z, \theta)$ is convex in $\theta$.*

**Assumption 2** (Surrogate Loss Regularity). *The surrogate loss $\ell(z, \theta)$ is continuously thrice differentiable in $\theta$, and the first three derivatives are bounded by $M_\ell$ in Frobenius norm.*

Consider any iterative algorithm (e.g., random sampling or uncertainty sampling) that at each iteration $t$ adds a single point $z^{(t)}$ and minimizes the regularized training loss:

$$L_t(\theta) \overset{\text{def}}{=} \sum_{i=1}^{t} \ell(z^{(i)}, \theta) + \lambda\|\theta\|_2^2 \tag{5}$$

to produce $\theta_t$. Since $L_{t-1}$ and $L_t$ differ by only one point, we expect $\theta_{t-1}$ and $\theta_t$ to also be close. We can make this formal using Taylor's theorem. First, since $\theta_t$ is a minimizer, we have $\nabla L_t(\theta_t) = 0$. Then, since the loss is continuously twice-differentiable, for some $\theta'$ between $\theta_{t-1}$ and $\theta_t$:

$$0 = \nabla L_t(\theta_t) = \nabla L_t(\theta_{t-1}) + \nabla^2 L_t(\theta')(\theta_t - \theta_{t-1}). \tag{6}$$

Since $\ell$ is convex and the quadratic regularizer, $\nabla^2 L_t$ is invertible and we can solve for $\theta_t$:

$$\theta_t = \theta_{t-1} - [\nabla^2 L_t(\theta')]^{-1} \nabla L_t(\theta_{t-1}). \tag{7}$$

Since $\theta_{t-1}$ minimizes $L_{t-1}$, we have $\nabla L_{t-1}(\theta_{t-1}) = 0$. Also note that $L_t(\theta) = L_{t-1}(\theta) + \ell(z^{(t)}, \theta)$. Thus,

$$\theta_t = \theta_{t-1} - [\nabla^2 L_t(\theta')]^{-1} \nabla \ell(z^{(t)}, \theta_{t-1}). \tag{8}$$

The update above holds for any choice of $z^{(t)}$, in particular, when $z^{(t)}$ is chosen by random sampling or uncertainty sampling. For random sampling, $z^{(t)} \sim p^*$, so we have

$$\mathbb{E}[\nabla \ell(z^{(t)}, \theta_{t-1})] = \nabla L(\theta_{t-1}), \tag{9}$$

from which one can interpret the iterates of random sampling as preconditioned SGD on the population surrogate loss $L$.

## 3.2   Parameter Updates of Uncertainty Sampling

Let us now turn to uncertainty sampling. Whereas random sampling is preconditioned SGD on the population *surrogate* loss $L$, we will now show that uncertainty sampling is preconditioned SGD on the population *zero-one* loss $Z$.

The very rough intuition is as follows: the gradient $\nabla Z(\theta)$ only depends on the density at the decision boundary corresponding to $\theta$ since points not at the decision boundary contribute zero gradient. Asymptotically, uncertainty sampling selects points close to the decision boundary defined by $\theta$ and points are selected proportional to the density.

Based on (8), we seek to understand $\mathbb{E}[\nabla \ell(z^{(t)}, \theta_{t-1})]$, where $z^{(t)}$ is chosen by uncertainty sampling. First, we must define some concepts. Each parameter vector $\theta$ defines a *decision boundary*:

**Definition 3** (Decision Boundary)**.**

$$B_\theta \overset{\text{def}}{=} \{x : S(x, \theta) = 0\}. \tag{10}$$

If $S(x, \theta)$ is differentiable with respect to $x$ and $\nabla_x S(x, \theta) \neq 0$ for all $x \in B_\theta$, then by the implicit function theorem, $B_\theta$ is a $(d - 1)$-dimensional differentiable manifold and has measure zero (see Proposition 10 in the Appendix). When this condition is satisfied, the decision boundary is well behaved, and $Z$ and uncertainty sampling has nice properties. For these reasons, denote the set of parameters that meet this condition as *regular parameters* $\Theta_{\text{regular}}$:

**Definition 4** (Regular Parameters)**.**

$$\Theta_{regular} \overset{\text{def}}{=} \{\theta : \forall x \in B_\theta, \nabla_x S(x, \theta) \neq 0\}. \tag{11}$$

For logistic regression with identity features ($\phi(x) = x$), $\nabla_x S(x, \theta) = \theta$, so the only point not in $\Theta_{\text{regular}}$ is $\theta = 0$. For logistic regression with quadratic features, $\theta \cdot \phi(x) = x^\top A x + b^\top x + c$ (the parameters are $A$, $b$, and $c$), parameters where $A$ is non-singular and $c \neq \frac{1}{4} b^\top A^{-1} b$ are in $\Theta_{\text{regular}}$. Thus, the parameters not in $\Theta_{\text{regular}}$ have measure zero.

Another important quantity is the probability density at the decision boundary. Before defining this, we first need to make two assumptions on the data distribution $p^*$ and an assumption that the score function is smooth.

**Assumption 5** (Smooth PDF). *$p^*$ has an smooth (all derivatives exist) probability density function.*

**Assumption 6** (Bounded Support). *The support of $p^*$ is bounded.*

**Assumption 7** (Smooth Score). *The score $S(x, \theta)$ is smooth, that is, all derivatives with respect to $x$ and $\theta$ exist.*

Recall that the decision boundary $B_\theta$ has measure zero for $\theta \in \Theta_{\text{regular}}$. Assumption 5 implies that there is zero probability mass on all decision boundaries corresponding to $\theta \in \Theta_{\text{regular}}$ ($\mathbb{P}(x \in B_\theta) = 0$ for $x \sim p^*$). However, we can define a probability *density* on the decision boundary $B_\theta$:

$$b(\theta) \overset{\text{def}}{=} \lim_{h \to 0} \frac{\mathbb{P}(S(x, \theta) \leq h) - \mathbb{P}(S(x, \theta) \leq 0)}{h}. \tag{12}$$

If Assumptions 5, 6, and 7 hold, then the limit exists for $\theta \in \Theta_{\text{regular}}$. For this statement, see Proposition 12 in the appendix.

Now that we have defined the set of regular parameters $\Theta_{\text{regular}}$, the density at the decision boundary $b(\theta)$, and Assumptions 5, 6, and 7, we are ready to formally state the expected gradient of the surrogate loss on a point chosen by uncertainty sampling. Our main result, Theorem 8, states that as the minipool size goes to infinity, $\mathbb{E}[\nabla \ell(z^{(t)}, \theta)]$ tends in the direction of the gradient of the population zero-one loss $Z(\theta)$, where the expectation is with respect to the randomness of the minipool $\mathcal{X}_{\text{M}}$ and label $y$. In particular, let $z^{(t)}$ be chosen via uncertainty sampling with the parameters $\theta$: $x^{(t)} = \arg\min_{x \in \mathcal{X}_{\text{M}}} |S(x, \theta)|$ and $y^{(t)} \sim p^*(y \mid x^{(t)})$. We require that the size of the minipool goes to infinity (and thus the size of the unlabeled pool must go to infinity as well) to ensure that we are choosing points arbitrarily close to the decision boundary.

**Theorem 8** (Expected Uncertainty Sampling Gradient). *If Assumptions 2, 5, 6, and 7 hold and $\theta \in \Theta_{regular}$ and $b(\theta) \neq 0$, then if $z^{(t)}$ is chosen via uncertainty sampling with parameters $\theta$,*

$$\lim_{n_{minipool} \to \infty} \mathbb{E}[\nabla \ell(z^{(t)}, \theta)] = \frac{-\psi'(0)}{b(\theta)} \nabla Z(\theta). \tag{13}$$

Thus, similar to how random sampling yields preconditioned SGD on the population surrogate loss $L$ (9), uncertainty sampling yields preconditioned SGD on the population zero-one loss $Z$.

## 3.3 Descent Direction

So far, we have shown that uncertainty sampling is preconditioned SGD on the population zero-one loss $Z$ by analyzing $\mathbb{E}[\nabla \ell(z^{(t)}, \theta)]$. To show that these updates are descent directions on $Z$, we need to also consider the preconditioner $[\nabla^2 L_t(\theta')]^{-1}$ appearing in (8). Due to quadratic regularization (5), the preconditioner is positive definite. However, we need to be careful since the preconditioner depends on the iterate $\theta_t$ both through $\theta'$ and the function $L_t$. Because of this, we only move in a descent direction in expectation if $\|\nabla Z(\theta_{t-1})\| \geq \epsilon$ and for large enough regularization, which ensures that the dependence on $\theta_t$ doesn't change the preconditioner too much.

**Theorem 9** (Uncertainty Sampling Descent Direction). *Assume that Assumptions 1, 2, 5, 6, and 7 hold, and assume $\psi'(0) < 0$. For any $b_0 > 0$, $\epsilon > 0$, and $n$, for any sufficiently large $\lambda \geq 2M_\ell^{3/2} b_0^{1/2} (-\psi'(0))^{-1/2} \epsilon^{-1/2} n^{2/3}$, for all iterates of uncertainty sampling $\{\theta_t\}$ where $\theta_{t-1} \in \Theta_{regular}$, $\|\nabla Z(\theta_{t-1})\| \geq \epsilon$, and $b(\theta_{t-1}) \leq b_0$, as $n_{minipool} \to \infty$,*

$$\nabla Z(\theta_{t-1}) \cdot \mathbb{E}[\theta_t - \theta_{t-1} | \theta_{t-1}] < 0. \tag{14}$$

Although $\lambda$ may appear to have to be quite large, note that typical regularization is proportion to the number of data points, while this regularization can be sub-linear which corresponds to rather weak regularization for large $n$.

This result explains why uncertainty sampling can achieve lower zero-one loss than random sampling; because it is implicitly descending on $Z$. Further, since $Z$ is non-convex, uncertainty sampling can converge to different values depending on the initialization.

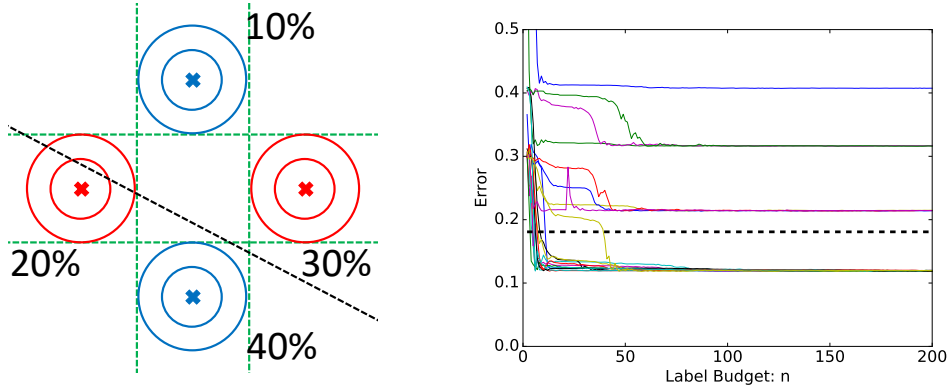

Figure 2: Synthetic dataset based on a mixture of four Gaussians (left) and the associated learning curves for runs of uncertainty sampling with different initial seed sets (right). Depending on the seed set, uncertainty sampling can produce either better or worse parameters than random sampling. See the main text for more information.

# 4 Experiments

We run uncertainty sampling on a simple synthetic data to illustrate the dynamics (Section 4.1) as well as 25 real datasets (Section 4.2). In both cases, we show how uncertainty sampling converges to different parameters depending on initialization, and how it can achieve lower asymptotic zero-one loss compared to minimizing the surrogate loss on all the data. Note that most active learning experiments are interested in measuring the rate of convergence (data efficiency), whereas this paper focuses exclusively on asymptotic values and the variation that we obtain from different seed sets. Also note that we measure only zero-one loss (error) but all algorithms optimize the logistic loss.

## 4.1 Synthetic Data

Figure 2 (left) shows a mixture of Gaussian distributions in two dimensions. All the Gaussians are isotropic, and the size of the circle indicates the variance (one standard deviation for the inner circle, and two standard deviations for the outer circle). The points drawn from the two red Gaussian distributions are labeled $y = 1$ and the points drawn from the two blue ones are labeled $y = -1$. The percentages refer to the mixture proportions of the clusters. We see that there are four local minima of the population zero-one loss, indicated by the green dashed lines. Each minima will misclassify one of the Gaussian clusters, yielding losses of just over 10%, 20%, 30%, and 40%. The black dotted line corresponds to the parameters that minimize the logistic loss, which yields a loss of 18%.

Figure 2 (right) shows learning curves for different seed sets, which consist of two points, one from each class. We see that the uncertainty sampling learning curves converge to four different asymptotic losses, corresponding to the four local minima of the zero-one loss mentioned earlier. The thick black dashed line is the zero-one loss for random sampling. We see that uncertainty sampling can achieve lower loss than random sampling. This occurs when the conditional label distribution is misspecified in a way that the (global) optimum of the logistic loss does not correspond to the global minimum of the zero-one loss.

## 4.2 Real-World Datasets

We collected 25 datasets from OpenML (retrieved August, 2017) that had a large number of data points and where logistic regression outperformed the majority classifier (predict the majority label). We further subsampled each dataset to have 10,000 points, which was divided into 7000 training points and 3000 test points. We ran a different version of uncertainty sampling from the version that we analyzed theoretically. Selecting from large minipools would require too much data and sampling without replacement (as is usually done in practice) would converge trivially since the entire dataset would eventually be labeled. Thus, instead of selecting points from the minipool subset and without replacement, we select points from the entire pool with replacement. We ran uncertainty sampling on

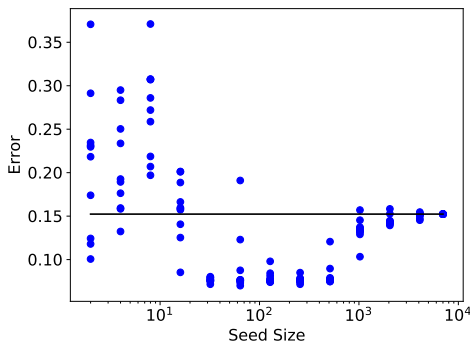 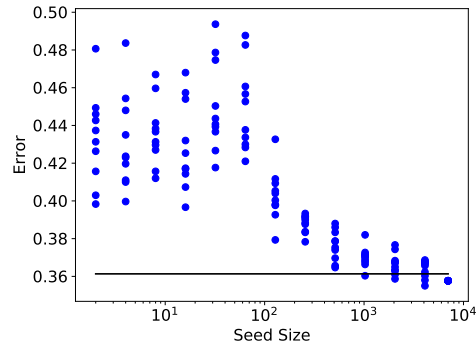

Figure 3: A scatter plot of the asymptotic zero-one loss for uncertainty sampling for two particular datasets for 13 seed sizes. The black line is the zero-one loss on the full dataset.

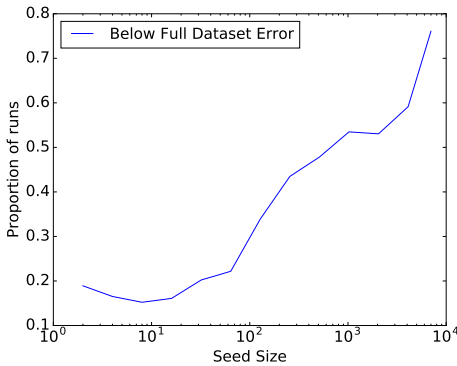 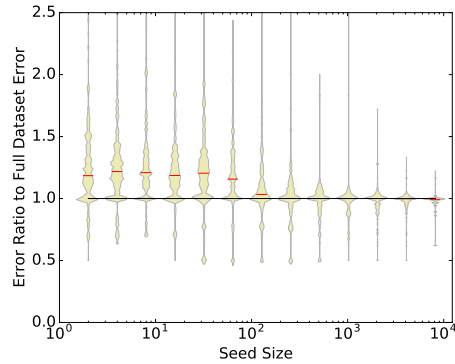

Figure 4: A plot showing the distribution of runs over the datasets (with 10 runs per dataset) of when uncertainty sampling converges to a lower zero-one loss than using the entire dataset.

Figure 5: A violin plot capturing the relative asymptotic zero-one loss compared to the zero-one loss on the full dataset. The plot shows the density of points with kernel density estimation. The red lines are the median losses. Each "violin" captures 230 points (10 runs over 23 datasets).

each dataset with random seed sets of sizes that are powers of two from 2 to 4096 and then 7000. We stopped when uncertainty sampling did not choose an unlabeled point for 1000 iterations. For each dataset and seed set size, we ran uncertainty sampling 10 times, for a total of $25 \cdot 13 \cdot 10 = 3250$ runs.

In Figure 3, we see scatter plots of the asymptotic zero-one loss of 130 points: 13 seed set sizes, each with 10 runs. The dataset on the left was chosen to exhibit the wide range of convergence values of uncertainty sampling, some with lower zero-one loss than with the full dataset. In both plots, we see that the variance of the zero-one loss of uncertainty sampling decreases as the seed set grows. This is expected from theory since the initialization has less variance for larger seed set sizes (as the seed set size goes to infinity, the parameters converge). For most of the datasets, the behavior was more similar to the plot on the right, where uncertainty sampling has a higher mean zero-one loss than random sampling for most seed sizes.

To gain a more quantitative understanding of all the datasets, we summarized the asymptotic zero-one loss of uncertainty sampling for various random seed set sizes. In Figure 4, we show the proportions of the runs over the datasets where uncertainty sampling converges to a lower zero-one loss than using the entire dataset. In Figure 5, we show a "violin plot" for the distribution of the ratio between the asymptotic zero-one loss of uncertainty sampling and the zero-one loss using the full dataset. We note that the mean and variance of uncertainty sampling significantly drops as the size of the seed set grows larger. The initial parameters are poor if the seed set is small, and it is well-known that poor initializations for optimizing non-convex functions locally can yield poor results, as seen here.

# 5 Related Work and Discussion

The phenomenon that uncertainty sampling can achieve lower error with a subset of the data rather than using the entire dataset has been observed multiple times. In fact, the original uncertainty sampling paper (Lewis and Gale, 1994) notes that "For 6 of 10 categories, the mean [F-score] for a classifier trained on a uncertainty sample of 999 examples actually exceeds that from training on the full training set of 319,463". Schohn and Cohn (2000) defines a heuristic that selects the point closest to the decision boundary of an SVM, which is equivalent to uncertainty sampling in our formulation. In the abstract, the authors note, "We observe... that a SVM trained on a well-chosen subset of the available corpus frequency performs better than one trained on *all* available data". More recently, Chang *et al.* (2017) develops an "active bias" technique that emphasizes the uncertain points and find that it increases the performance compared to using a fully-labeled dataset.

There is also work showing the bias of active learning can harm final performance. Schütze *et al.* (2006) notes the "missed cluster effect", where active learning can ignore clusters in the data and never query points from there; corresponding to a local minimum of the zero-one loss. Dasgupta and Hsu (2008) has a section on the bias of uncertainty sampling and provides another example where uncertainty sampling fails due to sampling bias, which we can explain as convergence to a spurious local minimum of the zero-one loss. Bach (2007) and Beygelzimer *et al.* (2009) note this bias issue and propose different importance sampling schemes to re-weight points and correct for the bias.

In this work, we find that uncertainty sampling updates are preconditioned SGD steps on the population zero-one loss and move in descent directions for parameters that are not approximate stationary points. Note that this does not give any global optimality guarantees. In fact, for linear classifiers, it is NP-hard to optimize the training zero-one loss below $\frac{1}{2} - \epsilon$ (for any $\epsilon > 0$) even when there is a linear classifier that achieves just $\epsilon$ training zero-one loss (Feldman *et al.*, 2012).

One of the key questions in light of this work is when optimizing convex surrogate losses yield good zero-one losses. If the loss function corresponds to the negative log-likelihood of a well-specified model, then the zero-one loss $Z$ will have a local minimum at the parameters that optimize the log-likelihood. If the loss function is "classification-calibrated", Bartlett *et al.* (2006) shows that if the convex surrogate loss of the estimated parameters converges to the optimal convex surrogate loss, then the zero-one loss of the estimated parameters converges to the global minimum of the zero-one loss (Bayes error). This holds only for universal classifiers (Micchelli *et al.*, 2006), but in practice, these assumptions are unrealistic. For instance, several papers show how outliers and noise can cause linear classifiers learned on convex surrogate losses to suffer high zero-one loss (Nguyen and Sanner, 2013; Wu and Liu, 2007; Long and Servedio, 2010).

Other works connect active learning with optimization in rather different ways. Ramdas and Singh (2013) uses active learning as a subroutine to improve stochastic convex optimization. Guillory *et al.* (2009) shows how performing online active learning updates corresponds to online optimization updates of non-convex functions, more specifically, truncated convex losses. In this work, we analyze active learning with offline optimization and show the connection between uncertainty sampling and one particularly important non-convex loss, the zero-one loss.

In summary, our work is the first to show a connection between the zero-one loss and the commonly-used uncertainty sampling. This provides an explanation and understanding of the various empirical phenomena observed in the active learning literature. Uncertainty sampling simultaneously offers the hope of converging to lower error but the danger of converging to local minima (an issue that can possibly be avoided with larger seed sizes). We hope this connection can lead to improved active learning and optimization algorithms.

**Reproducibility.** The code, data, and experiments for this paper are available on the CodaLab platform at `https://worksheets.codalab.org/worksheets/0xf8dfe5bcc1dc408fb54b3cc15a5abce8/`.

**Acknowledgments.** This research was supported by an NSF Graduate Fellowship to the first author.

## Footnotes

[1]Preconditioned refers to multiplication of a symmetric positive semidefinite matrix to the (stochastic) gradient for (stochastic) gradient descent (Li, 2018; Klein *et al.*, 2011). It is often chosen to approximate the inverse Hessian.

[2]For binary classification, the uncertainty measures of highest entropy, smallest margin, and most uncertain Settles (2010) are equivalent.

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
