[Supplementary Material · neurips2018_appendix.pdf]

# 6 Appendix

## 6.1 Manifold

**Proposition 10.** *If $S(x,\theta)$ is differentiable with respect to $x$ and $\nabla_x S(x,\theta) \neq 0$ throughout $B_\theta$, $B_\theta$ is an $(d-1)$-dimensional differentiable manifold and has measure zero.*

*Proof.* For any point $b \in B_\theta$, since $\nabla_x S(x,\theta) \neq 0$, there is some direction where $\nabla_x S(x,\theta)$ is non-zero. By the implicit function theorem, this means that there is a differentiable mapping from a subset of $\mathbb{R}^{d-1}$ to a neighborhood of $b$ within $B_\theta$. Thus, $B_\theta$ is a $(d-1)$-dimensional differentiable manifold. Further, in $\mathbb{R}^d$, every open cover has a countable subcover. Thus, there is a countable family of local patches (with local differentiable charts). Since each local patch is a continuous mapping from a measure zero set $\mathbb{R}^{d-1}$, the local patches have measure zero. Since a countable union of measure zero sets has measure zero, $B_\theta$ has measure zero. $\qquad\square$

## 6.2 Important Lemma

**Lemma 11.** *Suppose $\theta \in \Theta_{regular}$ and Assumption 7 holds. If $g(x)$ is smooth and has bounded support,*

$$F(s) = \int_{S(x,\theta)<s} g(x)dx \tag{15}$$

*is smooth at $0$.*

*Proof.* For this proof, we rely heavily on the arguments in Hoveijn (2007)

Since $g(x)$ has bounded support, for $\|x\| \geq M_x$, $g(x) = 0$. Intuitively, this means we can define a function that is equal to $S(x,\theta)$ for $\|x\| \leq M_x$ and is a small value $\|x\| \geq M_x$ and mollify to make it smooth. More precisely, let $S_{min} = \min(-2, \min_{\|x\|<2M_x} S(x,\theta))$. Define $f(x)$ to be equal to $S(x,\theta)$ inside a ball of radius $2M_x$ and equal to $S_{min}$ outside. Then mollify the function between balls of radius $M_x$ and $2M_x$. If we shift the function by $S_{min}$, the function is smooth, always positive, and vanishes at infinity. Thus, it satisfies the Shifted class C functions of Definition 2 of Hoveijn (2007).

Then, we can examine the function

$$G(s) = \int_{-1<f(x)<s} g(x)dx, \tag{16}$$

which will have the same derivatives (if they exist) as $F(s)$ around $0$. Note that $S_{min} \leq -2 < -1$, so the integration between the level sets is well-defined.

$0$ is a regular value because $\theta \in \Theta_{regular}$. Further, we don't need the non-degeneracy conditions of Hoveijn (2007) because $\nabla_x S(x,\theta)$ is continuous (Assumption 7) on a compact set (the support of $g(x)$) and thus is bounded below. And thus, a neighborhood around $0$ are regular values.

We can use the flow box and diffeomorphism argument from Hoveijn (2007) to express the volume function as an integral with $h$ as the upper limit (see Proposition 7 of Hoveijn (2007)). While Hoveijn (2007) uses $1$ as the integrand, the same argument holds for $g(x)$ as the integrand, and we recover that since $g(x)$ is smooth, the integral is smooth.

$\qquad\square$

## 6.3 Decision Boundary Density

**Proposition 12.** *If $\theta \in \Theta_{regular}$ and Assumptions 5, 6 and 7 hold, then $b(\theta)$ exists.*

*Proof.* The existence of $b(\theta)$ will follow from Lemma 11.

Define

$$F(s) = \int_{S(x,\theta)<s} p^*(x)dx \tag{17}$$

$$= \Pr_{x \sim p^*}[S(x,\theta) < s] \tag{18}$$

then $b(\theta) = F'(0)$ which exists by Lemma 11.

$\square$

### 6.3.1 Gradient of Z

**Lemma 13.**

$$\nabla Z(\theta) = -\frac{1}{2}\lim_{s\to 0}\frac{1}{s}\int_{|S(x,\theta)|\le s}\nabla_\theta S(x,\theta)\mathbb{E}[y|x]p(x)dx \tag{19}$$

*Proof.* The model classifies correctly when $S(x,\theta)y > 0$ and classifies incorrectly when $S(x,\theta)y < 0$

$$\nabla Z(\theta)\cdot a = \lim_{h\to 0}\frac{1}{2h}(Z(\theta+ha)-Z(\theta-ha)) \tag{20}$$

$$= \lim_{h\to 0}\frac{1}{2h}[\int_{S(x,\theta+ha)>0}\Pr[y=-1|x]dp(x) + \int_{S(x,\theta+ha)<0}\Pr[y=1|x]dp(x)- \tag{21}$$

$$- \int_{S(x,\theta-ha)>0}\Pr[y=-1|x]dp(x) - \int_{S(x,\theta-ha)<0}\Pr[y=1|x]dp(x)] \tag{22}$$

$$= \lim_{h\to 0}\frac{1}{2h}[\int_{S(x,\theta+ha)<0,S(x,\theta-ha)<0}(\Pr[y=1|x]-\Pr[y=1|x])dp(x)+ \tag{23}$$

$$+ \int_{S(x,\theta+ha)>0,S(x,\theta-ha)<0}(\Pr[y=-1|x]-\Pr[y=1|x])dp(x)+ \tag{24}$$

$$+ \int_{S(x,\theta+ha)<0,S(x,\theta-ha)>0}(\Pr[y=1|x]-\Pr[y=-1|x])dp(x)+ \tag{25}$$

$$+ \int_{S(x,\theta+ha)>0,S(x,\theta-ha)>0}(\Pr[y=-1|x]-\Pr[y=-1|x])dp(x)] \tag{26}$$

$$= \lim_{h\to 0}\frac{1}{2h}[\int_{S(x,\theta+ha)<0,S(x,\theta-ha)>0}\mathbb{E}[y|x]dp(x) - \int_{S(x,\theta+ha)>0,S(x,\theta-ha)<0}\mathbb{E}[y|x]dp(x)] \tag{27}$$

Applying Taylor's theorem,

$$\nabla Z(\theta)\cdot a = \lim_{h\to 0}\frac{1}{2h}[\int_{|S(x,\theta)|<-ha\cdot\nabla_\theta S(x,\theta)+O(h^2)}\mathbb{E}[y|x]dp(x) - \int_{|S(x,\theta)|<ha\cdot\nabla_\theta S(x,\theta)+O(h^2)}\mathbb{E}[y|x]dp(x)] \tag{28}$$

Because $h \to 0$,

$$\nabla Z(\theta)\cdot a = \lim_{h\to 0}\frac{1}{2h}[\int_{|S(x,\theta)|<-ha\cdot\nabla_\theta S(x,\theta)}\mathbb{E}[y|x]dp(x) - \int_{|S(x,\theta)|<ha\cdot\nabla_\theta S(x,\theta)}\mathbb{E}[y|x]dp(x)] \tag{29}$$

$$\nabla Z(\theta) \cdot a = \lim_{h \to 0} \int_{|S(x,\theta)| < |ha \cdot \nabla_\theta S(x,\theta)|} \frac{-\text{sgn}(ha \cdot \nabla_\theta S(x,\theta))}{2h} \mathbb{E}[y|x]dp(x) \tag{30}$$

$$\nabla Z(\theta) \cdot a = -\frac{1}{2} \lim_{h \to 0} \int_{|S(x,\theta)| < |ha \cdot \nabla_\theta S(x,\theta)|} \frac{1}{|ha \cdot \nabla_\theta S(x,\theta)|} a \cdot \nabla_\theta S(x,\theta) \mathbb{E}[y|x]dp(x) \tag{31}$$

$$\nabla Z(\theta) \cdot a = -\frac{1}{2} \lim_{s \to 0} \int_{|S(x,\theta)| < s} \frac{1}{s} a \cdot \nabla_\theta S(x,\theta) \mathbb{E}[y|x]dp(x) \tag{32}$$

$$\nabla Z(\theta) \cdot a = a \cdot -\frac{1}{2} \lim_{s \to 0} \frac{1}{s} \int_{|S(x,\theta)| < s} \nabla_\theta S(x,\theta) \mathbb{E}[y|x]dp(x) \tag{33}$$

$$\tag{34}$$

And thus,

$$\nabla Z(\theta) = -\frac{1}{2} \lim_{s \to 0} \frac{1}{s} \int_{|S(x,\theta)| < s} \nabla_\theta S(x,\theta) \mathbb{E}[y|x]dp(x) \tag{35}$$

$\square$

## 6.4 Expected gradient of loss for uncertainty sampling

**Theorem 8.** *If Assumptions 2, 5, 6, and 7 hold and $\theta \in \Theta_{regular}$ and $b(\theta) \neq 0$, then if $z^{(t)}$ is chosen via uncertainty sampling with the parameters $\theta$,*

$$\lim_{n_{minipool} \to \infty} \mathbb{E}[\nabla \ell(z^{(t)}, \theta)] = \frac{-\psi'(0)}{b(\theta)} \nabla Z(\theta). \tag{36}$$

*Proof.* We can decompose drawing the closest point as first drawing an absolute value of the score $s_2$ that is the *second closest* to 0 and then drawing the closest point conditioned on that score, which will be according to $p^*(x,y)$ among the $x$ with $|S(x,\theta)| \leq s_2$.

Let $r(s) = \mathbb{E}_{|S(x,\theta)| \leq s}[\mathbb{E}_{y|x}[\nabla_\theta \ell((x,y),\theta)]]$. As long as $s > 0$ and $P(|S(x,\theta)| \leq s) > 0$, it is well-defined quantity since $\nabla_\theta \ell(z^{(t)}; \theta) < M_\ell$. However, if $P(|S(x,\theta)| \leq s) = 0$ for $s > 0$, then $b(\theta) = 0$ (which we assumed is not the case). Thus, for $s > 0$, $r(s)$ is defined.

$$\lim_{n_{minipool} \to \infty} \mathbb{E}[\nabla \ell(z^{(t)}; \theta)] = \lim_{n_{minipool} \to \infty} \mathbb{E}[r(s_2)] \tag{37}$$

For any $s > 0$, $P(|S(x,\theta)| \leq s) > 0$ (from above) which implies that as $n_{minipool} \to \infty$, $P(s_2 \geq s) \to 0$. Thus,

$$s_2 \to_P 0 \tag{38}$$

Thus, since $\nabla_\theta \ell(z^{(t)}; \theta) < M_\ell$, $r(s_2)$ is bounded, so if the limit $\lim_{s \to 0} r(s)$ exists, then:

$$\lim_{n_{minipool} \to \infty} \mathbb{E}[r(s_2)] = \lim_{s \to 0} r(s) \tag{39}$$

$$\lim_{s \to 0} r(s) = \lim_{s \to 0} \mathbb{E}_{|S(x,\theta)| \leq s}[\nabla_\theta \ell(z, \theta)] \tag{40}$$

$$= \lim_{s \to 0} \frac{\int_{|S(x,\theta)| \leq s} \nabla_\theta \ell((x,y), \theta) dp^*(x,y)}{\int_{|S(x,\theta)| \leq s} dp^*(x,y)} \tag{41}$$

$$= \lim_{s \to 0} \frac{\int_{|S(x,\theta)| \leq s} \nabla_\theta \psi(yS(x,\theta)) dp^*(x,y)}{\int_{|S(x,\theta)| \leq s} dp^*(x,y)} \tag{42}$$

$$= \lim_{s \to 0} \frac{\int_{|S(x,\theta)| \leq s} \psi'(yS(x,\theta)) y \nabla_\theta S(x,\theta) dp^*(x,y)}{\int_{|S(x,\theta)| \leq s} dp^*(x,y)} \tag{43}$$

$$= \psi'(0) \lim_{s \to 0} \frac{\int_{|S(x,\theta)| \leq s} y \nabla_\theta S(x,\theta) dp^*(x,y)}{\int_{|S(x,\theta)| \leq s} dp^*(x,y)} \tag{44}$$

$$\lim_{s \to 0} r(s) = \psi'(0) \frac{\lim_{s \to 0} \frac{1}{s} \int_{|S(x,\theta)| \leq s} y \nabla_\theta S(x,\theta) dp^*(x,y)}{\lim_{s \to 0} \frac{1}{s} \int_{|S(x,\theta)| < s} p(x) dx} \tag{45}$$

The bottom limit exists by 12 and the top limit exists by an adaption of Proposition 12 with replacing the integrand $p^*(x)$ with $\nabla_\theta S(x,\theta)(p^*(x, y = 1) - p^*(x, y = -1))$ (which is smooth). This can be done by Lemma 11.

The bottom is exactly $2b(\theta)$,

$$\lim_{s \to 0} r(s) = \frac{\psi'(0)}{2b(\theta)} \lim_{s \to 0} \frac{1}{s} \int_{|S(x,\theta)| \leq s} y \nabla_\theta S(x,\theta) dp^*(x,y) \tag{46}$$

$$= \frac{-\psi'(0)}{b(\theta)} \left[ -\frac{1}{2} \lim_{s \to 0} \frac{1}{s} \int_{|S(x,\theta)| \leq s} y \nabla_\theta S(x,\theta) dp^*(x,y) \right] \tag{47}$$

$$= \frac{-\psi'(0)}{b(\theta)} \nabla Z(\theta) \tag{48}$$

The last line follows from Lemma 13. $\qquad\square$

## 6.5 Descent Direction

**Theorem 9.** *Assume that Assumptions 1, 2, 5, 6, and 7 hold, and assume $\psi'(0) < 0$. For any $b_0 > 0$, $\epsilon > 0$, and $n$, for any sufficiently large $\lambda \geq 2M_\ell^{3/2} b_0^{1/2} (-\psi'(0))^{-1/2} \epsilon^{-1/2} n^{2/3}$, for all iterates of uncertainty sampling $\{\theta_t\}$ where $\theta_{t-1} \in \Theta_{regular}$, $\|\nabla Z(\theta_{t-1})\| \geq \epsilon$, and $b(\theta_{t-1}) \leq b_0$, as $n_{minipool} \to \infty$,*

$$\nabla Z(\theta_{t-1}) \cdot \mathbb{E}[\theta_t - \theta_{t-1} | \theta_{t-1}] < 0. \tag{49}$$

*Proof.* The first thing to note is that if $\|\nabla Z(\theta_{t-1})\| > 0$, then $b(\theta_{t-1}) > 0$.

$$\|\nabla Z(\theta_{t-1})\| > 0 \tag{50}$$

$$\| -\frac{1}{2} \lim_{s \to 0} \frac{1}{s} \int_{|S(x,\theta_{t-1})| \leq s} \nabla S(x,\theta_{t-1}) \mathbb{E}[y|x] p(x) dx \| > 0 \tag{51}$$

$$\lim_{s \to 0} \frac{1}{s} \int_{|S(x,\theta_{t-1})| \leq s} \|\nabla S(x,\theta_{t-1})\| \|\mathbb{E}[y|x]\| p(x) dx > 0 \tag{52}$$

$$\lim_{s \to 0} \frac{1}{s} \int_{|S(x,\theta_{t-1})| \leq s} M_\ell p(x) dx > 0 \tag{53}$$

$$M_\ell b(\theta_{t-1}) > 0 \tag{54}$$

$$b(\theta_{t-1}) > 0 \tag{55}$$

$$\tag{56}$$

This will allow us to use Theorem 8 later in the proof.

As in the main text, we have

$$L_t(\theta) = \sum_{i=1}^{t} \ell(z^{(i)}, \theta) + \lambda \|\theta\|_2^2 \tag{57}$$

Thus, $L_t(\theta) = L_{t-1}(\theta) + \ell(z^{(t)}, \theta)$ and further $\nabla L_t(\theta_t) = 0$. Together, this implies that $\nabla L_t(\theta_{t-1}) = \nabla \ell(z^{(t)}, \theta_{t-1})$.

Using the Taylor expansion, for some value $\theta'$ on the line segment between $\theta_t$ and $\theta_{t-1}$,

$$0 = \nabla L_t(\theta_t) = \nabla \ell(z^{(t)}, \theta_{t-1}) + \nabla^2 L_t(\theta')(\theta_t - \theta_{t-1}) \tag{58}$$

$$\theta_t - \theta_{t-1} = -[\nabla^2 L_t(\theta')]^{-1} \nabla \ell(z^{(t)}, \theta_{t-1}) \tag{59}$$

$$\|\theta_t - \theta_{t-1}\| \le \frac{M_\ell}{\lambda} \tag{60}$$

Further, we can do another larger Taylor expansion,

$$0 = \nabla L_t(\theta_t) = \nabla \ell(z^{(t)}, \theta_{t-1}) + \nabla^2 L_t(\theta_{t-1})(\theta_t - \theta_{t-1}) + Q \tag{61}$$

where

$$Q_i = (\theta_t - \theta_{t-1})^T [\nabla^3 L_t(\theta'')]_i (\theta_t - \theta_{t-1}) \tag{62}$$

$$|Q_i| \le \frac{M_\ell}{\lambda} \|[\nabla^3 L_t(\theta'')]_i\|_F \frac{M_\ell}{\lambda} \tag{63}$$

$$\|Q\| \le \frac{M_\ell^3 n}{\lambda^2} \tag{64}$$

For simplicity, define $g = \nabla Z(\theta_{t-1})$.

From the three-term Taylor expansion,

$$\theta_t - \theta_{t-1} = -[\nabla^2 L_t(\theta_{t-1})]^{-1}(\nabla \ell(z^{(t)}, \theta_{t-1}) + Q) \tag{65}$$

$$-g \cdot (\theta_t - \theta_{t-1}) = g^T [\nabla^2 L_t(\theta_{t-1})]^{-1}(\nabla \ell(z^{(t)}, \theta_{t-1}) + Q) \tag{66}$$

$$= g^T [\nabla^2 L_t(\theta_{t-1})]^{-1} \nabla \ell(z^{(t)}, \theta_{t-1}) + g^T [\nabla^2 L_t(\theta_{t-1})]^{-1} Q \tag{67}$$

$$\ge g^T [\nabla^2 L_t(\theta_{t-1})]^{-1} \nabla \ell(z^{(t)}, \theta_{t-1}) - \|g\| \frac{1}{\lambda} \frac{M_\ell^3 n}{\lambda^2} \tag{68}$$

$$\ge g^T [\nabla^2 L_t(\theta_{t-1})]^{-1} \nabla \ell(z^{(t)}, \theta_{t-1}) - \frac{\|g\| M_\ell^3 n}{\lambda^3} \tag{69}$$

Noting that $(A + B)^{-1} = A^{-1} - A^{-1}B(A + B)^{-1}$, we can expand

$$[\nabla^2 L_t(\theta_{t-1})]^{-1} = [\nabla^2 L_{t-1}(\theta_{t-1})]^{-1} - R \tag{70}$$

where $R = [\nabla^2 L_{t-1}(\theta_{t-1})]^{-1} \nabla^2 \ell(z^{(t)}, \theta_{t-1})[\nabla^2 L_t(\theta_{t-1})]^{-1}$ and thus $\|R\| \le \frac{M_\ell}{\lambda^2}$

$$-g \cdot (\theta_t - \theta_{t-1}) \ge g^T [\nabla^2 L_{t-1}(\theta_{t-1})]^{-1} \ell(z^{(t)}, \theta_{t-1}) - \frac{\|g\| M_\ell^2}{\lambda^2} - \frac{\|g\| M_\ell^3 n}{\lambda^3} \tag{71}$$

On the right side, the only thing that depends on the randomness at iteration $t$ is $\ell(z^{(t)}, \theta_{t-1})$ whose expectation is given by Theorem 8 (this is where we use that $\theta \in \Theta_{\text{regular}}$ and $b(\theta) > 0$). So taking the expectation for uncertainty sampling and noting $n_{\text{minipool}} \to \infty$,

$$-g \cdot \mathbb{E}[\theta_t - \theta_{t-1}|\theta_{t-1}] \geq g^T[\nabla^2 L_{t-1}(\theta_{t-1})]^{-1}\frac{-\psi'(0)}{b(\theta_{t-1})}g - \frac{\|g\|M_\ell^2}{\lambda^2} - \frac{\|g\|M_\ell^3 n}{\lambda^3} \tag{72}$$

$$\geq \frac{-\psi'(0)}{b(\theta_{t-1})}\frac{\|g\|^2}{(t-1)M_\ell} - \frac{\|g\|M_\ell^2}{\lambda^2} - \frac{\|g\|M_\ell^3 n}{\lambda^3} \tag{73}$$

$$\geq \frac{-\psi'(0)}{b(\theta_{t-1})}\frac{\|g\|^2}{nM_\ell} - \frac{\|g\|M_\ell^2}{\lambda^2} - \frac{\|g\|M_\ell^3 n}{\lambda^3} \tag{74}$$

$$\geq \frac{\|g\|}{M_\ell n}\left[\frac{-\psi'(0)}{b(\theta_{t-1})}\|g\| - \frac{M_\ell^3 n}{\lambda^2} - \frac{M_\ell^4 n^2}{\lambda^3}\right] \tag{75}$$

$$\geq \frac{\epsilon}{M_\ell n}\left[\frac{-\psi'(0)}{b_0}\epsilon - \frac{M_\ell^3 n}{\lambda^2} - \frac{M_\ell^4 n^2}{\lambda^3}\right] \tag{76}$$

$$\tag{77}$$

Therefore, for $\lambda \geq 2M_\ell^{3/2} b_0^{1/2}(-\psi'(0))^{-1/2}\epsilon^{-1/2}n^{2/3}$ (and ensuring each power is at least 1),

$$-g \cdot \mathbb{E}[\theta_t - \theta_{t-1}|\theta_{t-1}] > 0 \tag{78}$$

Flipping the sign and plugging in $g$, we get the result.

$\square$