[Reviews · NeurIPS 2018]

Reviewer 1



This paper provides theoretical analysis and empirical examples for two phenomenon in active learning. The first is it could be possible that the 0-1 loss on subset of the entire dataset generated uncertainty sampling is smaller than learning using the whole dataset. The second is uncertainty sampling could “converge” to different models and predictive results. In the analysis, it is shown that the reason for these is the expected gradient of the “surrogate” loss of the most uncertain point is in the direction of the gradient of the current 0-1 loss. This result is based on the setup that the most uncertain point is sampled from a minipool that is a subset sampled without replacement randomly from the entire set. The result is proven in the limit when size of this minipool goes to infinity. In experiments, the two phenomenon is reproduced in both synthetic data and real world data, with slight change of the setup, for example, uncertain points are sampled with replacement there. My major concern of this paper is whether the theoretical analysis can explain the two wide observed phenomenon in active learning. The main result about 0-1 loss may be interesting. But it only says about the descent direction on the loss with current parameters in each iteration, not the true loss on the whole dataset. So the explanation that we are getting better 0-1 loss with subsets because we are minimizing actual 0-1 loss, not surrogate log loss or hinge loss, is not valid for me. Maybe I am missing some points. But at least there is something missing there in the paper. Actually, there could be various reasons for the first phenomenon. For example, we may have some noise that is far away from the decision boundary in the dataset and uncertainty sampling could end up with a subset that is “cleaner” than the whole set. Another issue is the experiments on real data to try different seed size. The setup is sampling the most uncertain points with replacement, which is different with the theoretical analysis and a rare case in active learning. I believe otherwise (without replacement) the variance would be much smaller, all converging to the same decision boundary. It is well known that with smaller random seeds, the active learner is easy to get stuck in wrong decision boundaries due to sampling bias. I am basically not convinced that it is due to multiple solutions of optimizing non-convex loss. The convergence in this paper refers to the case when sampling data with replacement that the active learner does not query new data, which should be defined earlier in the paper.

Reviewer 2



The paper shows that after some reasonable assumptions are made, a modified version of uncertainty sampling could be viewed as optimizing zero-one loss of a binary classification problem. Before reading this paper, I also wondered why some papers report that only looking at a subset of samples could improve the classifier performance. I am glad to see a theoretical paper which tries to answer this mystery. I believe that the results will inspire more people to develop better active learning or optimization algorithms. Strength: Except the typos, the authors seem to pay attention to subtle details in the proof. Some proving techniques they use are pretty clever, although I did not read enough theoretical papers to know how difficult you can come up with those techniques. It is impressive that the main result (theorem 9) does not require the convex assumption of the model. Except the proposition 11 and 12, which sound true but its proofs involve too advanced calculus concepts, I tried to follow every step in the proof. I think all steps are intuitively correct (after ignoring those typos) and are not redundant, although I did not derive every step by myself again to make sure it is 100% right. Weakness: The main weakness is its writing. First, the author misses some directly related work such as [1] and [2]. [1] and [2] also show the connections between active learning and gradient descent, although they prove the properties of different algorithms and different models. Second, the paper has too many typos and inconsistent notations. In addition, the authors often use terminologies without citations or explanations. These make the readers hard to follow the math, especially the proofs. Third, the theorem actually requires many assumptions, which might not hold in practice and are not stated very clearly in the abstract and introduction. For example, the abstract does not mention that the theorem only holds when large regularization is applied to the classifier and uncertainty sampling selects the points on the decision boundary (as the pool of samples approach to infinity and the probability mass near decision boundary is not zero). Typos and revision suggestions In the main paper: 22: At the first time of mentioning pre-condition SGD, providing a citation because I think not all people know what pre-condition means. 76: four parameters? Equation (5): saying that you are using Mean-Value theorem 152: Could you define smooth more clearly? 157 and 168: Theta_{valid} should be Theta_{regular} Equation (14): It took me some time to find what psi(x) is at line 50. Recommend to add the definition again here. In supplementary materials: Equation (17): There is no n_{seed} in (4) Equation (20): B -> M_l Equation (23): I think n should be removed, but equation (24) is correct Equation (31): Missing ) after theta_{t-1}. Missing gradient sign before l(z^(t),theta_{t-1}) Equation (39): Remove extra ] for E Equation (43): Missing E_{y|x} Equation (47) and (48): phi() -> psi() [1] Guillory, Andrew, Erick Chastain, and Jeff Bilmes. "Active learning as non-convex optimization." Artificial Intelligence and Statistics. 2009. [2] Ramdas, Aaditya, and Aarti Singh. "Algorithmic connections between active learning and stochastic convex optimization." International Conference on Algorithmic Learning Theory. Springer, Berlin, Heidelberg, 2013. After rebuttal: I think the authors address the concerns of reviewer 1 effectively. I encourage the authors to add that convergence proof to supplementary material. I keep my original judgment because I think all reviewers did not find significant flaws other than typos in the proof and I think the proposed theorems are significant enough to be published in NIPS.

Reviewer 3



This paper shows that uncertainty sampling is performing preconditioned stochastic gradient descent on the expected zero-one loss, which provides some explanation of the empirical finding that active learning with uncertainty sampling could yield lower zero-one loss than passive learning even with fewer labeled examples. Experiments on synthetic and real data are performed to illustrate and confirm some theoretical findings. Overall, the theory in the paper seems interesting, in particular Theorem 9 and the two supporting lemmas 13 and 14 that derive the gradient of the expected zero-one loss, which seems somewhat surprising because the empirical zero-one loss is not continuous. The experiments on synthetic data provide a nice and clear illustration of how uncertainty sampling may help to converge to lower zero-one loss, but the experiments on the real data seem less enlightening. The technical details seem mostly correct, although some typos are present and further clarification is needed in some places. In particular, in the proof of Lemma 14, it is not clear why O(h^2) in equation (61) can be ignored simply because h approaches 0. In the region where S(x, \theta) changes by O(h^2), couldn't the probability mass still be big, say o(h)? Minor comments on notations and typos: * In equation (8), it seems more clear to first define F_t to be the conditional expectation conditioned on \theta_{t-1}, and then say in the case of random sampling, F_t equals the gradient of L at \theta_{t-1}. * In Assumption 8, what exactly does being smooth mean? * In the statement of Theorem 9, what is \Theta_{valid}? Is it \Theta_{regular}? What is \psi?

Reviewer 4



This paper shows that under technical conditions, as the size of the "minipool" approaches infinity, uncertainty sampling is moving in the descent directions of the expected 0-1 loss in expectation. Therefore, uncertainty sampling may be interpreted as performing SGD with respect to the expected 0-1 loss. Based on this interpretation, the authors tried explaining why uncertainty sampling sometimes yields a smaller risk than the standard random sampling approach and why different initialization gives different outputs, as these are common behaviors of the SGD. The authors' perspective is quite fresh and inspiring, but there are some weakness. First, the main result, Corollary 10, is not very strong. It is asymptotic, and requires the iterates to lie in a "good" set of regular parameters; the condition on the iterates was not checked. Corollary 10 only requires a lower bound on the regularization parameter; however, if the parameter is set too large such that the regularization term is dominating, then the output will be statistically meaningless. Second, there is an obvious gap between the interpretation and what has been proved. Even if Corollary 10 holds under more general and acceptable conditions, it only says that uncertainty sampling iterates along the descent directions of the expected 0-1 loss. I don't think that one may claim that uncertainty sampling is SGD merely based on Corollary 10. Furthermore, existing results for SGD require some regularity conditions on the objective function, and the learning rate should be chosen properly with respect to the conditions; as the conditions were not checked for the expected 0-1 loss and the "learning rate" in uncertainty sampling was not specified, it seems not very rigorous to explain empirical observations based on existing results of SGD. The paper is overall well-structured. I appreciate the authors' trying providing some intuitive explanations of the proofs, though there are some over-simplifications in my view. The writing looks very hasty; there are many typos and minor grammar mistakes. I would say that this work is a good starting point for an interesting research direction, but currently not very sufficient for publication. Other comments: 1. ln. 52: Not all convex programs can be efficiently solved. See, e.g. "Gradient methods for minimizing composite functions" by Yu. Nesterov. 2. ln. 55: I don't see why the regularized empirical risk minimizer will converge to the risk minimizer without any condition on, for example, the regularization parameter. 3. ln. 180--182: Corollar 10 only shows that uncertainty sampling moves in descent directions of the expected 0-1 loss; this does not necessarily mean that uncertainty sampling is not minimizing the expected convex surrogate. 4. ln. 182--184: Non-convexity may not be an issue for the SGD to converge, if the function Z has some good properties. 5. The proofs in the supplementary material are too terse.